# 2,5-Bis(2,2,2-trifluoroethoxy)phenyl-tethered 1,3,4-Oxadiazoles Derivatives: Synthesis, In Silico Studies, and Biological Assessment as Potential Candidates for Anti-Cancer and Anti-Diabetic Agent

**DOI:** 10.3390/molecules27248694

**Published:** 2022-12-08

**Authors:** Sathyanarayana D. Shankara, Arun M. Isloor, Avinash K. Kudva, Shamprasad Varija Raghu, Pavan K. Jayaswamy, Pushyaraga P. Venugopal, Praveenkumar Shetty, Debashree Chakraborty

**Affiliations:** 1Membrane and Separation Technology Laboratory, Department of Chemistry, National Institute of Technology Karnataka, Surathkal, Mangalore 575025, India; 2Solara Active Pharma Sciences, No:120 A&B, Industrial Area, Baikampady, New Mangalore, Mangalore 575011, India; 3Department of Biochemistry, Mangalore University, Mangalagangothri, Mangalore 574199, India; 4Neurogenetics Lab, Department of Applied Zoology, Mangalore University, Mangalagangothri, Mangalore 574199, India; 5Central Research Laboratory, KS. Hegde Medical Academy, Nitte (Deemed to be University), Deralakatte, Mangalore 575018, India; 6Biophysical and Computational Chemistry Laboratory, Department of Chemistry, National Institute of Technology Karnataka, Surathkal, Mangalore 575025, India; 7Department of Biochemistry, K.S. Hegde Medical Academy, Nitte (Deemed to be University), Deralakatte, Mangalore 575018, India

**Keywords:** 1,3,4-oxadiazole, anti-cancer studies, *Glioblastoma*, anti-diabetic studies, docking simulation studies, *Drosophila*

## Abstract

In the present work, a series of new 1-{5-[2,5-bis(2,2,2-trifluoroethoxy)phenyl]-1,3,4-oxadiazol-3-acetyl-2-aryl-2H/methyl derivatives were synthesized through a multistep reaction sequence. The compounds were synthesized by the condensation of various aldehydes and acetophenones with the laboratory-synthesized acid hydrazide, which afforded the Schiff’s bases. Cyclization of the Schiff bases yielded 1,3,4-oxadiazole derivatives. By spectral analysis, the structures of the newly synthesized compounds were elucidated, and further, their anti-cancer and anti-diabetic properties were investigated. To examine the dynamic behavior of the candidates at the binding site of the protein, molecular docking experiments on the synthesized compounds were performed, followed by a molecular dynamic simulation. ADMET (chemical absorption, distribution, metabolism, excretion, and toxicity) prediction revealed that most of the synthesized compounds follow Lipinski’s rule of 5. The results were further correlated with biological studies. Using a cytotoxic assay, the newly synthesized 1,3,4-Oxadiazoles were screened for their in vitro cytotoxic efficacy against the LN229 *Glioblastoma* cell line. From the cytotoxic assay, the compounds **5b**, **5d**, and **5m** were taken for colony formation assay and tunnel assay have shown significant cell apoptosis by damaging the DNA of cancer cells. The in vivo studies using a genetically modified diabetic model, *Drosophila melanogaster,* indicated that compounds **5d** and **5f** have better anti-diabetic activity among the different synthesized compounds. These compounds lowered the glucose levels significantly in the tested model.

## 1. Introduction

In the group of heteroaromatic cyclic compounds, mainly heterocycles containing nitrogen, oxygen, and sulfur atom in the ring are considered to display enormous application in pharmacology, industries, polymer science, and as agrochemicals [1,2,3,4,5,6]. Over the past few periods, significant evidence has been gathered to evince the effectualness of oxadiazoles in the field of pharmaceutical chemistry. The combinatorial drugs in the field of novel drug discovery research play a significant role by forming hybrid drugs with improved efficiency and multi-targeting properties [2,7,8]. Among them, 1,3,4-oxadiazole compounds earn significant value in the heterocyclic class because of their wide range of biological applications [6,9,10]. 

The 1,3,4-Oxadiazole is a five-membered heteroaromatic molecule, and it is a thermally stable neutral system. The 1,3,4-Oxadiazoles have a vast variety of applications in the areas of medicine, agriculture, dyes, heat-resistant polymers, scintillators, UV absorbing, and fluorescent materials [11]. These potential bioactive agents also show a variety of biological significance such as anticancer [12,13,14], antimicrobial [15], anti-diabetic [16,17,18], antibacterial [19], antifungal [20], antiviral [21], anticonvulsant [22], anti-inflammatory [23], analgesic [24], and insecticidal [25] activities, and thus have drawn widespread attention. 

Fluorine-containing compounds have been of great significance in the field of pharmaceuticals because of their various pharmacological actions. The introduction of this electronegative element in the form of trifluoromethyl, an alternative to the hydrogen atom, was observed to enhance its activity when matched to the non-fluorine molecules [26]. The aspects such as solubility, absorption, metabolism, bioavailability, and excretion responsible for better activity increase in the presence of fluorine atom in a molecule [27]. The more lipophilic nature of fluorine atom than hydrogen will intensify the fat-soluble and hydrogen bond-accepting nature of the fluorine-introduced molecule [28].

Cancer is the second major threat to the human body after heart disease. It develops normally when healthy cells in a particular body part grow uncontrollably. The principle involved in the formation of different types of cancers are the same; they grow, divide, and redivide and form new abnormal cells [12,29]. As its leading cause of death, the treatment of cancer has become a prime endeavor of research and development because of several side effects and drug resistance [30]. The most prevalent and deadly kind of brain cancer is *glioma* or *glioblastoma* (GBM). Moreover, it is the most malignant type of primary brain tumor of the glial cells, delineated by high invasiveness, resistance to chemotherapy, and a great risk of recurrence [31,32,33,34,35]. The symptom appears and progresses rapidly, and it is characterized by a poor prognosis and a largely incurable one. The drug temozolomide is being employed for *glioblastoma* therapy, generally concomitant with radiotherapy [36]. A drawback of temozolomide is that it increases the expression of O6-methylguanine DNA methyltransferase (MGMT) and decreases the ability of GBM cells to repair DNA [37]. Therefore, there is a dire need for targeted drug delivery to control *glioblastoma* cells.

Diabetes mellitus, characterized by high glucose levels is a chronic metabolic disorder with several different types of causes, mainly resistance to insulin, lack of insulin, and defects in the gene of the β-cells of the pancreas. The pancreatic hormone insulin plays a major role in maintaining the blood glucose level in the body. The lower level of insulin production in the body leads to excess deposition of glucose in the blood, which results in diabetic conditions [38]. The non-insulin-dependent Diabetes Mellitus type 2 (DMT2) is the most widespread non-communicable disease, and it affects more than 85–90% of diabetic patients [39,40]. DMT2 is characterized by an intense degree of metabolic miscarriage and a shoot-up of blood glucose levels [41,42]. To medicate and control DMT2, the main factor is to reduce hyperglycemia with the help of α-glucosidase inhibitors [43]. Inquisition of the literature reveals that five-membered 1,3,4-oxadiazole derivatives showed prominent anti-hyperglycemic activity [16,17,18].

The fruit fly, *Drosophila melanogaster*, is widely employed as an animal model to study different biological questions. Mammals and *Drosophila* share many physiological and biological properties in common. In addition, there is about 75% sequence similarity for disease genes between *Drosophila* and humans. At room temperature, they have a short life cycle of 10–12 days, and the easy maintenance in the lab has prompted the usage of *Drosophila* as an in vivo model. *Drosophila* offers many transgenic tools to study and investigate genetic effects [44,45]. In the present study, genetically modified diabetic *Drosophila melanogaster* flies were used to evaluate the anti-diabetic properties of 1,3,4-Oxadiazole derivatives.

The above observations prompted us to synthesize 1,3,4-oxadiazole containing a trifluoro group as possible anticancer and antidiabetic molecules. The interaction between proteins and ligands has been discovered using an in silico method. The anti-cancer and anti-diabetic properties were evaluated using in vitro and in vivo biological models, respectively. 

## 2. Results and Discussion

### 2.1. Chemistry

The sequence of reactions involved in the synthesis of said title compounds is given in Figure 1. The synthesis of 1,3,4-oxadiazole derivatives has been made in the following two ways: one using different aldehydes, and the other one using various acetophenones.

The key intermediate, 2,5-bis(2,2,2-trifluoroethoxy)benzohydrazide (**3**), was obtained by the esterification product of 2,5-bis(2,2,2-trifluoroethoxy)benzoic acid. Through the condensation of various aldehydes/acetophenones with 2,5-bis(2,2,2-trifluoroethoxy)benzohydrazide, the various hydrazone derivatives (**4a–n**) were synthesized. The prepared hydrazones were further refluxed with acetic anhydride, yielding 1-{5-[2,5-bis(2,2,2-trifluoroethoxy)phenyl]-1,3,4-oxadiazol-3(2*H*)-yl}ethenone (**5a–h**) and 1-{5-[2,5-bis(2,2,2-trifluoroethoxy)phenyl]-1,3,4-oxadiazol-3(2 methyl)-yl}ethenone (**5i–n**). As stated in the experimental section, IR, ^1^H-NMR, ^13^C-NMR, and mass spectrophotometry analyses were performed to determine the structures of the newly synthesized compounds.

In the IR spectrum of 1-{5-[2,5-bis(2,2,2-trifluoroethoxy)phenyl]-2-(2,6-dichlorophenyl)-1,3,4-oxadiazol-3(2*H*)-yl}ethanone (**5a**), The band for ArC-H was found at 2956 cm^−1^. The band for the CH group appeared at 2918 cm^−1^. The band at 1655 cm^−1^ for carbonyl absorption. At 1460 cm^−1^, the -C=C stretch was noticed. At 1261 cm^−1^, the -C-O-C stretch was found. 

Furthermore, in the ^1^H NMR spectrum of compound **5a**, the trifluoro ethoxy group attached to the parent moiety’s CH_2_ group appeared as a multiplet at frequencies between δ 4.3 and δ 4.5, which are suitable for four protons. At δ 2.31, protons of the acetyl group of the oxadiazole ring have appeared as a singlet. The oxadiazole C-5 proton is responsible for the singlet signal at δ 7.49. The signals from the 2,6-dichloro phenyl moiety’s three aromatic protons overlapped and resonated as multiplets in the range of δ 7.01–7.09. While the 3H and 4H protons entered resonance at δ 7.31–7.35, the 6H proton of the parent phenyl moiety first emerged as a singlet at δ 7.23. Similarly, in the ^13^C NMR spectrum of the compound, the acetyl CH_3_ gave a peak at δ 21.1. The CH_2_ of the trifluoroethoxy group is at δ 66.2–68.4. The CF_3_ couplings were observed at δ 118–126 and the carbonyl carbon gave a peak of δ 167.6. Moreover, the mass spectrum showed a molecular ion peak at *m*/*z* 533 (M + 1 peak), which is consistent with its molecular formula, C_20_H_14_Cl_2_F_6_N_2_O_4_. 

The structural data of the newly synthesized molecules are presented in Table 1.

### 2.2. Pharmacology

#### 2.2.1. Molecular Docking Studies

The information regarding the interaction between a drug molecule and a protein was provided by molecular docking. 

Using the same docking procedure, the 1,3,4-oxadiazole derivatives were docked to the AURKA (Aurora kinase A) and VEGFR-2 (Vascular Endothelial Growth Factor Receptor-2) proteins shown in Figure 1a,b, respectively. It was anticipated that drug compounds would be screened based on their stable binding poses and related binding energies. Docking studies revealed that hydrophobic and π-stacking interactions are the primary determinants of the interactions. All the molecules showed fewer tendencies to form hydrogen bonding interactions with both AURKA and VEGFR-2. Similarly, the 1,3,4-oxadiazole derivatives were docked to the α-glucosidase binding pocket to predict the stable binding pose and affinity shown in Figure 1c. The binding affinity of the 1,3,4-oxadiazole derivatives obtained from docking studies is given in Table 2. The docking of 1,3,4-oxadiazole derivatives with AURKA and VEGFR-2 revealed the highest binding affinity in the range −10.1–−8.5 kcal/mol and −9.5–−8.8 kcal/mol, respectively, as compared to standard drug Temozolomide (−5.7 kcal/mol, −5.9 kcal/mol). Similarly, 1,3,4-oxadiazole derivatives showed the highest affinity range of −9.2–−7.9 kcal/mol towards α-Glucosidase compared to acarbose (−7.4 kcal/mol), an anti-diabetic drug. 

**AURKA:** The non-covalent interactions between kinase and the drug candidates are mainly located in the region of Gly 140 to Asp 274 amino acid residues, which cover the binding pocket of the kinase. The derivatives mainly form hydrophobic interactions with the residues Leu 139, Val 147, Leu 149, Ala 160, Leu 263, Leu 164, Leu 194, Ala 213, and Ala 273. The compounds, **5b**, **5c,** and **5g** formed a strong hydrogen bond with Lys 162 residue at a distance of 1.94 Å, 1.97 Å, and 2.23 Å, respectively. The compounds **5f** and **7i** formed a relatively weak hydrogen bond with Lys 141 residue at a distance of 2.42 Å, 2.61 Å, and 2.29 Å, respectively. The π-π stacking interactions shown by 1,3,4-oxadiazole derivatives are given in Table 3. The distance characteristics for π-π—stacking interactions are 4.5–5.5 and the angle between the normal vectors of two rings is 0–90. [46]. It can be seen that the molecules prefer to form edge-to-face interactions with the amino acid residues of AURKA at a stable distance of around 5–6 Å than face-to-face interactions. 

The docking results were further confirmed by a 30 ns molecular dynamics simulation. The most active compound was selected, **5b**, to study the dynamic behavior of the ligands at the protein binding sites. Figure 2a shows the RMSD (root mean square deviation) profile of the ligand and backbone C_α_ atoms about the protein. It is observed that the protein fluctuations are stable in the complex, which indicates that ligand binding does not change the conformation of residues at the binding pocket. The movement of **5b** copes well with the kinase backbone, which suggests that the complex of **5b** showed greater stability at the binding pocket. 

Similar to π-π interactions, hydrogen bond interactions were found to stabilize the ligand at the binding pocket. An average of a single hydrogen bond interaction is stable throughout the 30 ns simulation, which supports the results from molecular docking (Figure 3a). A detailed analysis of interactions showed the formation of hydrogen bonds between various residues, which are not obtained from docking studies, which are due to the ability of the ligand to attain favorable conformation at the pocket. 

**VEGFR-2:** The interactions found between VEGFR-2 and drug candidates were mainly hydrophobic in nature, as in the case of kinase enzymes. The interactions mainly reside in the area of amino acid residues from His 816 to Leu 1049, which covers the binding site of the receptor. The amino acid residues Cys 817, Ala 866, Val 848, Ala 881, Leu 882, Ile 888, Leu 889, Ile 892, Val 898, Val 916, Leu 1019, Cys 1024, Ile 1025, and Leu 1049 mainly showed hydrophobic interactions with the bulky groups in the molecule. Among these derivatives, only **5b** showed a hydrogen bonding interaction with residue Arg 1027 (3.25 Å) and Ile 1025 (2.15 Å), respectively. 

The π-π stacking interactions shown by 1,3,4-oxadiazole derivatives with VEGFR-2 are given in Table 4. It is found that the majority of oxadiazole derivatives form stable edge-to-face interactions with the residue His1026 at a distance of around 5.8–6.5 Å. Other compounds tend to form weak face-to-face interactions with aromatic residues at the binding pockets.

The compound **5b** showed three hydrogen bond interactions with Ala 213, Tyr 212, and Arg 137, where Ala 213 has a higher contribution to the stability (Table 5).

A 25 ns MD simulation was carried out for the complex of the most active compound, **5b**, with VEGFR-2. The RMSD profile of the backbone C_α_ atoms of VEGFR-2 and its ligand with respect to proteins were illustrated in Figure 2b. The movement of **5b** copes well with the VEGFR-2 backbone after 10 ns, which suggests that the complex of **5b** showed greater stability at the binding pocket. Similarly, the protein fluctuations became stable after 10 ns of the simulation, which indicates that the ligand binding does not change the conformation of the binding pocket. The simulation showed that hydrogen bond interactions stabilize the ligand at the binding pocket. An average of single hydrogen bond interactions is stable throughout the simulation trajectory (Figure 3b). A detailed analysis showed the formation of hydrogen bonds between Asp1046, Ile1025, and His816, which are not obtained from docking studies. That can be caused by the ligand’s capacity to achieve a favorable conformation at the pocket.

**α-Glucosidase:** The binding affinity of 1,3,4-oxadiazole derivatives with α-Glucosidase is given in Table 2. The complex of compounds **5b**, **5c,** and **5d** was found to have a similar binding mode as acarbose, while the other derivatives showed a diverse binding mode and location. All three compounds showed π-π interaction with Trp 406 residue. Additionally, the complex of **5c** was found to have π-π interaction with Phe 450 residue. The complexes of **5b** and **5d** showed hydrogen bond interaction with Tyr 605 residue. The other compounds showed diverse binding sites, which are mainly stabilized by hydrophobic interactions rather than electrostatic interactions. The complexes formed at these binding sites are energetically less favorable as compared to the acarbose binding site. 

The results obtained from docking and experiments suggest that compound **5d** has a higher binding affinity among all the 1,3,4-oxadiazole derivatives. This is further confirmed by a short (20 ns) molecular dynamic simulation. The RMSD profile of α-Glucosidase backbone C_α_ atoms and ligand with respect to α-Glucosidase were given in Figure 4. It is observed that the protein fluctuations are stable in the complex, which indicates that ligand binding does not alter the conformation of amino acids at the binding sites. The movement of **5d** copes well with the α-Glucosidase backbone up to 14 ns. The compound showed higher fluctuation around 15 ns and moved out of the binding pocket, which again got synchronized with the binding pocket after 16 ns of the simulation. An average of a single hydrogen bond is observed throughout the 16 ns simulation, which supports the results from docking (Figure 5). Additional four hydrogen bonds are observed in the complex with the residues Gln 603 (1.7%), Lys 345 (5.4%), and Tyr299 (6.3%) due to the dynamic behavior of the compound at the ligand binding site.


**ADME Toxicity Prediction**


Using the Swiss ADME web tool, the pharmacokinetic (PK) parameters such as adsorption, distribution, metabolism, excretion, and toxicity were defined [47]. The drug-likeness, bioavailability, and effectiveness of the synthesized molecules were predicted using a set of protocols as given in Table 6. All the compounds obeyed Lipinski’s rule of five except for **5a, 5c, 5k,** and **5m**. Most of the compounds have molecular weight < 500 g/mol, favored for oral absorption. The compounds that have >500 g/mol MW indicate that these compounds get absorbed through membranes. The dipole moment, bonds that are rotatable, number of hydrogen bond acceptors, and number of hydrogen bond donors for all the molecules were found to be in the permissible range of 1.0–12.5 debye, 0–15, 0.0–6.0, and 2.0–20.0, respectively. The cell permeability value for all the compounds i.e., the polar surface area lies between 60.36 and 138.99 Å^2^, which shows good oral availability. If the polar surface area is <60 Å^2^, the compounds do not penetrate the blood–brain barrier well, which is evident from the BBB determination. The predicted PSA values are less than 140 Å^2^, which suggests that the compounds have good intestinal absorption properties. The molar refractivity explains the electronic polarizability of individual ions and their interaction in solutions. The synthesized compounds have refractivity values between 98.56 and 134.44 m^3^/mol. Two compounds, namely, **5a** and **5c,** have refractivity values above the permissible range. The predicted ADMET properties suggest that the compounds, except **5a** and **5c**, can serve as effective drug candidates.

#### 2.2.2. Cytotoxic Studies

The cytotoxic properties of newly synthesized fluorine-containing 1,3,4-Oxadiazolines derivatives on LN229 cells were studied using an MTT assay. Cells were treated with a concentration of 0, 5, 10, 15, 20, and 25 µm for a period of 48 h. 

The % cell viability data for the all-screened compounds are shown in Figure 6. The compounds (**5b**), (**5d**), and (**5m**) were screened with significant cytotoxic effects with 48.4%, 49.8%, and 46.3% of cell viability, respectively. The outcome pointed out that **5b** containing the 6-methoxy naphthyl group remarkably increased the cytotoxicity. On the contrary, the **5c** compound containing a biphenyl substituent has a less cytotoxic effect. The insertion of the chloro group into the fourth position at the substituent group of the aldehyde-derived oxadiazole molecule has shown outstanding anticancer properties among all synthesized molecules and the acetophenone-derived **5l** and **5n** have exposed the least activity. The second and sixth positions containing the chloro group of dichloroacetophenone derived **5m** has displayed excellent cytotoxicity by causing cancer cell death. While **5a**, **5e**, **5i**, and **5k** rendered moderate activity. The overall good activity of the molecules is mainly because of the bis trifluoro ethoxy group present at the parent moiety. Using absorbance values from the MTT assay, a non-linear regression analysis was used to measure the concentration of these compounds required to cause a 50% reduction in cell viability (IC_50_) as shown in Table 7. The compounds 5b, 5d, and 5m show IC_50_ concentrations such as 10.14 µM with 51.58% cell death, 8.141 µM with 50.12% cell death, and 10.48 µM with 53.65% cell death, respectively. The IC_50_ concentration is significant because it shows how much of a drug is required to block a biological process by 50%. Because of an increase in cell mortality or a decrease in cell growth, the IC curves demonstrate a shift in population. The representative IC_50_ graphs and % cell viability for the screened compounds are shown in Figure 7.

Further, to validate the efficacy of the selected compounds from MTT, such as **5b**, **5d**, and **5m,** were treated with a minimum of 10 µM and a maximum of 25 µM in the colony formation assay. All these compounds showed a significant cytotoxic effect at 25 µM and limited the ability of the single cells to form multiple colonies, as shown in Figure 8. In addition, these results are further validated by a tunnel assay, the principle of which relies on the detection of the OH-group of the DNA fragment formed during apoptosis. Moreover, these selected drugs showed significant anticancer properties as they are treated to LN229 cells by showcasing green fluorescence (Figure 9) implying that most of the cell’s DNA is damaged and the OH-group of the fragmented DNA is exposed to the specific fluorophore or reporter enzyme-labeled antibodies. Here, the compounds **5b**, **5d**, and **5m** were treated to glioblastoma cell lines for about 48 h to check the molecular efficacy of the screened compounds. Extensive fragmentation of nuclear DNA generates DNA strand breaks (DSBs), which were detected with green (FITC) labeled fluorochromes in treatment groups. Whereas, in positive control groups cells, were treated with the DNase enzyme, whose DNA tends to show high CFTC expression. In contrast, the negative control shows no fluoresces as their DNA was unaffected. The relative FITC expression of the screened compounds was analyzed with respect to the negative control. These results confirm the molecular frame of the screened compounds (**5b**, **5d**, and **5m**) by inducing cell apoptosis and showing them to have potential anti-cancer properties. 

#### 2.2.3. In Vivo Drosophila Study

The molecular docking analyses predict seven synthesized compounds, namely, **5b**, **5c**, **5d**, **5e**, **5f**, **5g**, and **5k** to bind the α-Glucosidase enzyme (Table 2). These compounds were further analyzed for their anti-diabetic properties using the *Drosophila* in vivo model. 

The estimation of glucose level in the hemolymph homogenate showed that **5d** and **5f** have lowered the glucose level in the diabetic fly model compared to the control. The diabetic flies showed higher glucose levels compared to the wild-type control flies. The compound **5d** contains electron-withdrawing groups such as 4-chloro at the para position of phenyl, and **5f** contains chloro at the ortho position of 1,3,4-oxadiazole. The molecule possessing the hydroxy group **5g** has the least activity. The larger molecules containing napthalyl (**5b**) and biphenyl (**5c**) have moderate inhibition towards glucosidase. The docking and the in vivo analysis clearly showed that the compounds **5d** and **5f** have a clear α-Glucosidase inhibitory activity. The possible inhibition of α-Glucosidase by the compounds **5d** and **5f** are responsible for lowering the glucose levels in the in vivo model. In vivo analysis of glucose level in control and the *Drosophila,* diabetic model is presented in Figure 10.

## 3. Conclusions

From a straightforward chemical procedure, the target, two series of new substituted 1,3,4-oxadiazole derivatives having trifluoro ethoxy moiety, were synthesized, using different aldehydes (**5a–h**) and other series by using various acetophenones (**5i–n**). The structural confirmation was performed by spectral studies and screened for their antidiabetic and anticancer properties.

The drug-protein interaction of synthesized compounds (**5a–g**) and (**5i–m**) for selected biological activities were performed by molecular docking studies. According to in silico study, synthesized oxadiazole compounds interacted with AURKA, VEGFR-2 of *glioblastoma* brain tumor, and α-glycosidase have higher binding affinity than the taken standard drug and may inhibit the activity of the enzymes AURKA, VEGFR-2, and glycosidase via hydrophobic and π-stacking interactions. The MD simulation of the most active compound with the enzymes revealed the stability of the protein–ligand complex and supported the results obtained from docking. Simulation results showed that hydrogen bond interactions are equally important for stabilizing ligands at the binding pocket. ADMET properties suggest that the compounds **5b**, **5d**, **5e**, **5f**, **5g**, **5i**, **5j,** and **5l** can serve as effective drug candidates.

The synthesized compounds containing the trifluoroethoxy group were further assessed for their in vitro cytotoxic properties against LN-229 *glioblastoma* cell lines. Likewise, compounds containing the electron-withdrawing 6-methoxy naphthyl group (**5b**), 4-chloro phenyl (**5d**), and 2,6-dichloro phenyl (**5m**), as a substituent of 1,3,4-oxadiazole derived from aldehydes and acetophenone, prompt to exhibit significant cytotoxic effects among the newly synthesized 1,3,4-oxadiazole derivatives from MTT assay. This was further confirmed by limiting the rate of colonies that are formed and, also, by inducing cellular apoptosis (tunel assay) at a 25 µm concentration. The presence of the electron-withdrawing group on the aromatic ring linked to the oxadiazole ring can considerably boost the anti-proliferative effect [48,49]. The assertion is also proven by previous studies that hybrids carrying electron-withdrawing groups improved the activity and F > Cl > Br > NO_2_ > CH_3_ was the order of electron-withdrawing groups in terms of potency [50].

The α-Glucosidase is a key enzyme that regulates the blood glucose level by specifically hydrolyzing 1,4-α-glucopyranoside bonds to produce α-glucose [51]. Previous studies have shown that, inhibiting α-Glucosidase activity will lead to reduced absorption of glucose and decrease the blood glucose level [52]. Thus α-Glucosidase is a key target for treating diabetes and developing α-Glucosidase inhibitors is essential for effective therapeutic intervention in diabetic patients. Our study on the anti-diabetic property using in vivo *Drosophila* model showed that, the compounds **5d** and **5f** bind to the α-Glucosidase enzyme. These compounds bearing an additional electron-withdrawing group such as the chloro-group at ortho and para positions on the phenyl ring (in **5d** and **5f**) seemed to have good enzyme inhibitory effects, which corroborated the previous report performed elsewhere [18]. Moreover, our in silico docking analysis also supports this claim (Table 2). Hence, these compounds may be further studied for the potential development of a novel class of antidiabetic agents.

To summarize, the strongly deactivating trifluoroethoxy group of oxadiazole moiety enhances the activity towards anticancer and antidiabetic. In which the substituent containing electron-withdrawing groups such as 6-methoxy naphthyl group (5b), 4-chloro phenyl (5d), 2,6-dichloro phenyl (5m), 2-chloro phenyl (5f), and 2,4-dichloro phenyl (5k) have shown excellent activity that warranted the more extensive study to prove it as a better candidate. Additionally, the 5d molecule may be considered a “hit compound” since it is potent toward both anticancer and antidiabetic activity and needed further optimization to make it a “lead compound”, as this showed overall high activity and multi-targeting ability. 

## 4. Materials and Methods

### 4.1. Chemistry

Open capillary analysis was followed to examine the Melting point, and the results were uncorrected. A Bruker FTIR-4100 spectrophotometer was used to record the IR spectra, which were made using the KBr pellet method. TMS was used as an internal standard during the ^1^H NMR analysis of the compounds in CDCl_3_ solvent using a Joel (400 MHz) instrument. Chemical shift values were expressed in ppm or δ units. On a Shimadzu lab solution spectrometer running at 70 eV, the mass spectra were captured. TLC was used to observe the progress of the reaction using aluminum sheets covered with silica gel (silica gel 60 F254) that were purchased from Merck. Commercial-grade solvents and reagents were utilized to purify generated chemicals.

### 4.2. Synthesis of 2,5-Bis(2,2,2-trifluoroethoxy)benzohydrazide (***3***)

The 2,5-bis(2,2,2-trifluoroethoxy)benzoate (**2**) was prepared from 2,5-bis(2,2,2-trifluoroethoxy)benzoic acid (0.03 mmol) by reaction with thionyl chloride (0.06 mmol) in methanol at reflux temperature. Reaction completion was monitored by TLC. Then distillation of mass to remove excess thionyl chloride, neutralization by sodium bicarbonate, and extraction using ethyl acetate. Furthermore, the concentration of extracted mass is to obtain esterification product. (Yield = 90%).

Further, absolute alcohol (25 mL) was used to dissolve 2,5-bis(2,2,2-trifluoroethoxy)benzoate (**2**) (0.003 mol) and 99% hydrazine hydrate (0.0031 mol), after which the clear solution was refluxed for three hours. TLC was performed to check reaction completion. After cooling, the material was distilled to remove extra alcohol. The resulting solid was filtered and using a little amount of cooled ethanol it was washed thoroughly, and finally dried at 55–60 °C. (Yield is 94%).

### 4.3. Synthesis of Hydrazones (***4a–n***)

In ethanol, the mixture of 2,5-bis(2,2,2-trifluoroethoxy)benzohydrazide, (**3**) (0.003 mol) and suitable aldehyde/acetophenone (0.0033 mol) was taken and heated to 70 °C for 2 h. After the reaction completion, the mass was slowly cooled to 25–30 °C and then to 0–5 °C. The precipitated solid was Buchner filtered and washed with ethanol that had been precooled and dried at 55–60 °C.

### 4.4. Procedure for the Synthesis of 1-{5-[2,5-Bis(2,2,2-trifluoroethoxy)phenyl]-1,3,4-oxadiazol-3(2H/CH_3_)-yl}ethanone (***5a–h***) and (***5i–n***)

A mixture of **4a–n** (0.01 mmol) mixed with acetic anhydride (10 mL) was heated to reflux for 8 h [53,54]. By examining TLC, the reaction’s completion was observed. Then the excess acetic anhydride was taken out by distillation at below 65 °C under vacuum and the resulting thick, syrupy mixture was then added to ice water. The solid separated after vigorous swirling. The obtained solid was filtered and rinsed with cold water and dried at 55–60 °C. The pure product obtained after recrystallization in hot ethanol solvent (**5a–h**)/(**5i–n**).

#### 4.4.1. 1-{5-[2,5-Bis(2,2,2-trifluoroethoxy)phenyl]-2-(2,6-dichlorophenyl)-1,3,4-oxadiazol-3(2H)-yl}ethanone (**5a**)

White microcrystals; m.p. 127–129 °C; IR (KBr) γ/cm^−1^: 2956 (Ar-C-H), 2918 (-C-H), 1655 (-C=O), 1609.28 (-C=N), 1261.3 (-C-O-C); ^1^H NMR (δ ppm, CDCl_3_, 400 MHz): 2.31 (s, 3H, -COCH_3_), 4.30–4.50 (m, 4H, -CH_2_), 7.01–7.09 (m, 3H, 2,6-dichloro phenyl CH), 7.23–7.35 (m, 3H, trifluoro ethoxyphenyl CH), 7.49 (s, 1H, oxadiazole 2H); ^13^C NMR (δ ppm, CDCl_3_, 100 MHz): δ 21.10 (-COCH_3_), δ 66.25–68.42 (CH_2_CF_3_), δ 86.04 (C_5_ of oxadiazol), δ 114.85,115.07,116.46 (C_3,4,5_ of substituent aromatic ring), δ 118.00–126.18(CF_3_-coupling), δ 131.61 (C_1_ of substituent aromatic ring), δ 134.93,134.88 (C_2,6_ of substituent aromatic ring), δ 126.21 (C_4_ of parent aromatic ring), δ 131.31 (C_3_ of parent aromatic ring), δ 151.68 (C_1_ of parent aromatic ring), δ 151.31 (C_6_ of parent aromatic ring), δ 152.84 (C_2_ of oxadiazol), δ 163.47, 160.94 (C_2,5_ of parent aromatic ring), 167.61 (C=O); LC-MS (*m*/*z*): 533 (M + 1), (molecular formula C_20_H_14_Cl_2_F_6_N_2_O_4_).

#### 4.4.2. 1-{5-[2,5-Bis(2,2,2-trifluoroethoxy)phenyl]-2-(6-methoxynapthalyl)-1,3,4-oxadiazol-3(2H)-yl}ethanone (**5b**)

White microcrystals; m.p. 126–128 °C; IR (KBr) γ/cm^−1^: 2971 (Ar-C-H), 2942 (-C-H), 1671 (-C=O), 1609.28 (-C=N), 1223.3 (C-O-C); ^1^H NMR (δ ppm, CDCl_3_, 400 MHz): 2.38 (s, 3H, -COCH_3_), 3.91 (s, 3H, -OCH_3_), 4.29–4.47 (m, 4H, -CH_2,_ J = 4.0Hz), 7.00–7.37 (m, 6H, naphthyl CH), 7.49–7.75 (m, 3H, trifluoro ethoxyphenyl CH, J = 8.8Hz) 7.9 (s, 1H, oxadiazole 2H); ^13^C NMR (δ ppm, CDCl_3_, 100 MHz): δ 21.39 (-COCH_3_), δ 55.30 (-OCH_3_), δ 66.26–68.21 (CH_2_CF_3_), δ 91.88 (C_5_ of oxadiazol), δ 105.66, 115.96, 116.43 (C_3,4,5_ of substituent naphthyl ring), δ 117.39–124.40 (CF_3_-coupling), δ 126.57, 127.74 (C_2,8_ of substituent naphthyl ring), δ 128.29 (C_6_ of substituent naphthyl ring), δ 135.43 (C_1_ of substituent naphthyl ring), δ 134.93, 134.88 (C_9,10_ of substituent naphthyl ring), δ 131.10 (C_6_ of parent aromatic ring), δ 129.96 (C_4_ of parent aromatic ring), δ 131.31 (C_3_ of parent aromatic ring), δ 151.29 (C_1_ of parent aromatic ring), δ 135.43 (C_6_ of parent aromatic ring), δ 152.74, 152.17 (C_2,5_ of parent aromatic ring), δ 158.47 (C_2_ of oxadiazol), 168.15 (C=O); LC-MS (*m*/*z*): 543 (M + 1) molecular formula = C_25_H_20_F_6_N_2_O_5._

#### 4.4.3. 1-{5-[2,5-Bis(2,2,2-trifluoroethoxy) phenyl]-2-(biphenyl)-1,3,4-oxadiazol-3(2H)-yl}ethanone (**5c**)

White microcrystals; m.p. 138–140 °C; IR (KBr) γ/cm^−1^: 3033 (Ar-C-H), 2942 (-C-H), 1668 (-C=O), 1609.28 (C=N), 1214.7 (-C-O-C); ^1^H NMR (δ ppm, CDCl_3_, 400 MHz): 2.38 (s, 3H, -COCH_3_), 4.314–4.494 (m, 4H,- CH_2_), 7.006–7.102 (m, 3H, trifluoro ethoxyphenyl CH), 7.34–7.61 (m, 9H, biphenyl CH), 7.63 (s, 1H, oxadiazole 2H); ^13^C NMR (δ ppm, CDCl_3_, 100 MHz): δ 21.27 (-COCH_3_), δ 66.19–68.26 (-CH_2_CF_3_), δ 84.33 (C_5_ of oxadiazol), δ 110.70, 110.78, 112.25 (C_9,10,11_ of substituent biphenyl ring), δ 114.98–119.51(CF_3_-coupling), δ 121.11, 121.35, 121.66, 124.49 (C_2,6,8,12_ of substituent biphenyl ring), δ 147.93, 148.05, 143.90 (C_1,4,7_ of substituent biphenyl ring), δ 150.32 (C_4_ of parent aromatic ring), δ 151.27 (C_3_ of parent aromatic ring), δ 152.74 (C_1_ of parent aromatic ring), δ 151.98 (C_6_ of parent aromatic ring), δ 152.97 (C_2_ of oxadiazol), δ 168.00, 165.99 (C_2,5_ of parent aromatic ring), 177.88 (-C=O) LC-MS (*m*/*z*): 539.15 (M + 1) molecular formula = C_26_H_20_F_6_N_2_O_4._

#### 4.4.4. 1-{5-[2,5-Bis(2,2,2-trifluoroethoxy)phenyl]-2-(4-chlorophenyl)-1,3,4-oxadiazol-3(2H)-yl}ethanone (**5d**) 

White microcrystals; m.p. 118–120 °C; IR (KBr) γ/cm^−1^: 3066.52 (Ar-C-H), 2970.28 (-C-H), 1612.40 (-C=O), 1521.50 (-C=N), 1219.20 (-C-O-C); ^1^H NMR (δ ppm, CDCl_3_, 400 MHz): 2.34 (s, 3H, -COCH_3_), 4.31–4.45 (m, 4H, -CH_2_), 6.99–7.10 (m, 3H, trifluoro ethoxyphenyl CH), 7.34–7.44(m, 4H, Ar-CH), 7.45 (s, 1H, oxadiazole 2H); ^13^C NMR (δ ppm, CDCl3, 100 MHz): δ 20.76 (-COCH_3_), δ 66.21–68.40 (-CH_2_CF_3_), δ 90.78 (C_5_ of oxadiazol), δ 119.53–129.04 (CF_3_-coupling), δ 114.97, 116.07, 116.14, 117.04 (C_2,3,5,6_ of substituent aromatic ring), δ 119.53–129.04 (CF_3_-coupling), δ 130.90 (C_1_ of substituent aromatic ring), δ 134.68 (C_4_ of substituent aromatic ring), δ 140.96 (C_4_ of parent aromatic ring), δ 150.33 (C_3_ of parent aromatic ring), δ 151.28 (C_1_ of parent aromatic ring), δ 152.17 (C_6_ of parent aromatic ring), δ 152.70, 153.01 (C_2,5_ of parent aromatic ring), δ 159.67 (C_2_ of oxadiazol), 168.20 (-C=O); LC-MS (*m*/*z*): 497.1 (M + 1) molecular formula = C_20_H_15_ClF_6_N_2_O_4._

#### 4.4.5. 1-{5-[2,5-Bis(2,2,2-trifluoroethoxy)phenyl]-2-(4-furfuryl)-1,3,4-oxadiazol-3(2H)-yl}ethenone (**5e**)

White microcrystals; m.p. 116–118 °C; IR (KBr) γ/cm^−1^: 3135.91 (Ar-C-H), 2946.92 (-C-H), 1659.2 (-C=O), 1612.28 (-C=N), 1217.3 (-C-O-C); ^1^H NMR (δ ppm, CDCl_3_, 400 MHz): 2.35 (s, 3H, -COCH_3_), 4.31–4.47 (m, 4H, -CH_2_), 6.39–6.4 (m, 1H, Furfuryl-CH), 6.62–6.62 (d, 1H, Furfuryl-CH), 6.99–7.08 (m, 3H, trifluoro ethoxyphenyl CH), 7.34–7.34 (d, 1H, Furfuryl-CH), 7.43 (s, 1H, oxadiazole 2H); ^13^C NMR (δ ppm, CDCl_3_, 100 MHz): δ 20.77 (-COCH_3_), δ 66.21–67.48 (CH_2_CF_3_), δ 114.95 (C_5_ of oxadiazol), δ 117.06–121.45 (CF_3_-coupling), δ 127.36 (C_4_ of parent aromatic ring), δ 127.68 (C_3_ of parent aromatic ring), δ 128.46 (C_6_ of parent aromatic ring), δ 129.01 (C_3_ of furan ring), δ 130.27 (C_4_ of furan ring), δ 135.16 (C_1_ of parent aromatic ring), δ 139.70 (C_5_ of furan ring), δ 147.20 (C_2_ of furan ring), δ 152.99, 150.33 (C_2,5_ of parent aromatic ring), δ 159.63 (C_2_ of oxadiazol), 165.85 (-C=O); LC-MS (*m*/*z*): 453.05 (M + 1) molecular formula = C_18_H_14_F_6_N_2_O_5._

#### 4.4.6. 1-{5-[2,5-Bis(2,2,2-trifluoroethoxy)phenyl]-2-(2-chlorophenyl)-1,3,4-oxadiazol-3(2H)-yl}ethenone (**5f**)

White microcrystals; m.p. 115–117 °C; IR (KBr) γ/cm^−1^: 3090.89 (Ar-C-H), 2892.65 (-C-H), 1660.87 (-C=O), 1606.17 (-C=N), 1227.66 (-C-O-C); ^1^H NMR (δ ppm, CDCl_3_, 400 MHz): 2.39 (s, 3H, -COCH_3_), 4.29–4.44 (m, 4H, -CH_2_), 6.98–7.00(m, 1H, Ar-CH), 7.04–7.07 (d, 1H, Ar-CH), 7.28–7.33 (m, 3H, trifluoro ethoxyphenyl CH), 7.35–7.39 (m, 2H, Ar-CH), 7.46 (s, 1H, oxadiazole 2H); ^13^C NMR (δ ppm, CDCl_3_, 100MHz): δ 21.26 (-COCH_3_), δ 66.28–68.24 (-CH_2_CF_3_), δ 89.525 (C_5_ of oxadiazol), δ 116.09, 116.33, 117.40 (CF_3_-coupling), δ 119.46, 121.65, 124.41, 127.26 (C_3,4,5,6_ of substituent aromatic ring), δ 128.46 (C_1_ of substituent aromatic ring), δ 130.38 (C_2_ of substituent aromatic ring), δ 131.22 (C_4_ of parent aromatic ring), δ 132.76 (C_3_ of parent aromatic ring), δ 133.53 (C_1_ of parent aromatic ring), δ 151.28 (C_2_ of oxadiazole ring), δ 152.71, 152.05 (C_2,5_ of parent aromatic ring), 168.07 (C=O); LC-MS (*m*/*z*): 496.9 (M + 1) molecular formula = C_20_H_15_ClF_6_N_2_O_4._

#### 4.4.7. 1-{5-[2,5-Bis(2,2,2-trifluoroethoxy)phenyl]-2-(3-flouro-4-methoxyphenyl)-1,3,4-oxadiazol-3(2-methyl)-yl}ethanone (**5i**)

White microcrystals; m.p. 90–92 °C; IR (KBr) γ/cm^−1^: 2944.91 (Ar-C-H), 2848.50 (-C-H), 1654.47 (-C=O), 1589.19 (-C=N), 1219.93 (-C-O-C); ^1^H NMR (δ ppm, CDCl_3_, 400 MHz): 2.24 (s, 3H, -COCH_3_), 2.32 (s, 3H,Oxadiazol -CH_3_), 3.89 (s, 3H, -OCH_3_), 4.32–4.44 (m, 4H, -CH_2_), 6.93–7.09 (m, 3H, Ar-CH), 7.26–7.35 (m, 3H, trifluoro ethoxyphenyl CH); ^13^C NMR (δ ppm, CDCl_3_, 100 MHz): δ 22.36 (CH_3_), δ 22.84 (-COCH_3_), δ 56.1 (OCH_3_), δ 66.31–68.05 (-CH_2_CF_3_), δ 99.1 (C_5_ of oxadiazol), δ 112.80, 113.93, 114.13 (C_2,5,6_ of substituent aromatic ring), δ 116.09, 116.47, 117.02 (CF_3_-coupling), δ 119.15 (C_1_ of substituent aromatic ring), δ 122.03 (C_4_ of parent aromatic ring), δ 124.47 (C_3_ of parent aromatic ring), δ 131.98 (C_1_ of parent aromatic ring), δ 148.20 (C_4_ of substituent aromatic ring), δ 150.36 (C_3_ of substituent aromatic ring), δ 152.68, 151.23 (C_2,5_ of parent aromatic ring), δ 153.19 (C_2_ of oxadiazol), δ 168.85 (-C=O); LC-MS (*m*/*z*): 525.15 (M + 1, 100%), molecular formula = C_22_H_19_F_7_N_2_O_5._

#### 4.4.8. 1-{5-[2,5-. bis(2,2,2-trifluoroethoxy)phenyl]-2-(4-hydroxy phenyl)-1,3,4-oxadiazol-3(2-methyl)-yl}ethanone (**5j**)

White microcrystals; m.p. 121–123 °C; IR (KBr) γ/cm^−1^: 3074.85 (Ar-C-H), 2897.04 (-C-H), 1666.73 (-C=O), 1579.57 (-C=N), 1221.95 (-C-O-C); ^1^H NMR (δ ppm, CDCl_3_, 400 MHz): 2.21 (s, 3H, -COCH_3_), 2.22 (s, 3H,Oxadiazol CH_3_), 4.26(s, 1H, -OH),4.28–4.37 (m, 4H, -CH_2_), 6.91–7.05 (m, 4H, Ar-CH), 7.27–7.53 (m, 3H, trifluoro ethoxyphenyl CH); ^13^C NMR (δ ppm, CDCl_3_, 100 MHz): δ 22.3(CH_3_), 23.1 (-COCH_3_), δ 65.90–68.32 (-CH_2_CF_3_), δ 99.40 (C_5_ of oxadiazol), δ 115.01, 115.88, 116.35, 117.02 (C_2,3,5,6_ of substituent aromatic ring), δ 119.28, 121.16, 121.25, 121.49, 121.72, 124.42 (CF_3_-coupling), δ 127.15 (C_4_ of parent aromatic ring), δ 129.88 (C_3_ of parent aromatic ring), δ 134.65 (C_1_ of substituent aromatic ring), δ 151.17 (C_1_ of parent aromatic ring), δ 150.28 (C_4_ of substituent aromatic ring), δ 152.63,152.91 (C_2,5_ of parent aromatic ring), δ 154.29 (C_2_ of oxadiazol), δ 169.22 (-C=O); LC-MS (*m*/*z*): 491.1 (M − 1), molecular formula = C_22_H_19_F_7_N_2_O_5._

#### 4.4.9. 1-{5-[2,5-Bis(2,2,2-trifluoroethoxy)phenyl]-2-(2,4-dichlorophenyl)-1,3,4-oxadiazol-3(2-methyl)-yl}ethenone (**5k**)

White microcrystals; m.p. 88–90 °C; IR (KBr) γ/cm^−1^: 3003.71 (Ar-C-H), 2944.40 (-C-H), 1663.50 (-C=O), 1591.39 (-C=N), 1216.59 (-C-O-C); ^1^H NMR (δ ppm, CDCl_3_, 400 MHz): 2.21 (s, 3H, -COCH_3_), 2.28 (s, 3H,Oxadiazol CH_3_), 4.30–4.48 (m, 4H, -CH_2_), 6.99–7.08 (m, 2H, Ar-CH), 7.30–7.43 (m, 3H, trifluoro ethoxyphenyl CH), 7.62–7.65 (d, 1 H, Ar-CH); ^13^C NMR (δ ppm, CDCl_3_, 100 MHz): δ 21.74 (CH_3_), δ 23.15 (-COCH_3_), δ 65.9–68.5 (-CH_2_CF_3_), δ 97.82 (C_5_ of oxadiazol)), δ 115.91, 116.62, 117.43 (C_3,5,6_ of substituent aromatic ring), δ 119.27, 121.66, 121.73, 124.43 (CF_3_-coupling), δ 124.49 (C_4_ of parent aromatic ring), δ 126.84 (C_3_ of parent aromatic ring), δ 130.51 (C_1_ of substituent aromatic ring), δ 133.32 (C_1_ of parent aromatic ring), δ 133.61 (C_4_ of substituent aromatic ring), δ 136.05 (C_2_ of substituent aromatic ring), δ 151.33, 150.58 (C_2,5_ of parent aromatic ring), δ 152.74 (C_2_ of oxadiazol), δ 167.04 (-C=O); LC-MS (*m*/*z*): 545 (M + 1), molecular formula = C_21_H_16_Cl_2_F_6_N_2_O_4._

#### 4.4.10. 1-{5-[2,5-Bis(2,2,2-trifluoroethoxy)phenyl]-2-(4-florophenyl)-1,3,4-oxadiazol-3(2-methyl)-yl}ethenone (**5l**)

White microcrystals; m.p. 67–69 °C; IR (KBr) γ/cm^−1^: 3018.81 (Ar-C-H), 2938 (-C-H), 1661.10 (-C=O), 1596.87 (-C=N), 1228. (-C-O-C); ^1^H NMR (δ ppm, CDCl_3_, 400 MHz): δ 2.26 (s, 3H, -COCH_3_), 2.31 (s, 3H,Oxadiazol CH_3_), 4.32–4.43 (m, 4H, -CH_2_), 6.97–7.08 (m, 4H, Ar-CH), 7.34–7.57 (m, 3H, trifluoro ethoxyphenyl CH); ^13^C NMR (δ ppm, CDCl_3_, 100 MHz): δ 22.37 (CH_3_), δ 23.05 (-COCH_3_), δ 66.33–68.03 (-CH_2_CF_3_), δ 99.42 (C_5_ of oxadiazol), δ 115.23, 115.45, 116.18, 116.48 (C_2,3,5,6_ of substituent aromatic ring), δ 119.28, 121.16, 121.25, 121.49, 121.72, 124.42 (CF_3_-coupling), δ 127.15 (C_4_ of parent aromatic ring), δ 129.88 (C_3_ of parent aromatic ring), δ 135.07 (C_1_ of substituent aromatic ring), δ 150.44 (C_1_ of parent aromatic ring), δ 151.24 (C_4_ of substituent aromatic ring), δ 164.28, 161.80 (C_2,5_ of parent aromatic ring), δ 152.96 (C_2_ of oxadiazol), δ 167.35 (-C=O); LC-MS (*m*/*z*): 493.05 (M − 1), molecular formula = C_21_H_17_ClF_6_N_2_O_4._

#### 4.4.11. 1-{5-[2,5-Bis(2,2,2-trifluoroethoxy)phenyl]-2-(2,6-dichlorophenyl)-1,3,4-oxadiazol-3(2-methyl)-yl}ethenone (**5m**)

White microcrystals; m.p. 94–96 °C; IR (KBr) γ/cm^−1^: 3078.46 (Ar-C-H), 1663.63 (-C=O), 1591.54 (-C=N), 1216.72 (-C-O-C); ^1^H NMR (δ ppm, CDCl_3_, 400 MHz): 2.13 (s, 3H, -COCH_3_), 2.21 (s, 3H,Oxadiazol CH_3_), δ 4.21–4.38 (m, 4H, -CH_2_), 6.91–7.26 (m, 3H, Ar-CH), 7.32–7.65 (m, 3H, trifluoro ethoxyphenyl CH); LC-MS (*m*/*z*): 545 (M + 1), molecular formula = C_21_H_16_Cl_2_F_6_N_2_O_4._

#### 4.4.12. 1-{5-[2,5-bis(2,2,2-trifluoroethoxy)phenyl]-2-(4-chlorophenyl)-1,3,4-oxadiazol-3(2-methyl)-yl}ethenone (**5n**)

White microcrystals; m.p. 106–108 °C; IR (KBr) γ/cm^−1^: 3090.89 (Ar-C-H), 2937 (-C-H), 1660.87 (-C=O), 1625.37 (-C=N), 1227.7 (-C-O-C); ^1^H NMR (δ ppm, CDCl_3_, 400 MHz): 2.18 (s, 3H, -COCH_3_), 2.24 (s, 3H,Oxadiazol CH_3_), 4.25–4.36 (m, 4H, -CH_2_), 6.90–6.93 (d, 1H, Ar-CH), 6.98–7.01 (d, 1H, Ar-CH), 7.27–7.29 (m, 3H, trifluoro ethoxyphenyl CH), 7.42–7.44 (m, 2H, Ar-CH); ^13^C NMR (δ ppm, CDCl_3_, 100 MHz): δ 22.36 (CH_3_), δ 22.84 (-COCH_3_), δ 66.32–67.98 (-CH_2_CF_3_), δ 99.33 (CH_3_ of oxadiazol), δ 116.15, 116.37, 116.91, 118.90 (C_2,3,5,6_ of substituent aromatic ring), δ 119.17, 121.66, 124.43 (CF_3_-coupling), δ 127.35 (C_4_ of parent aromatic ring), δ 128.62 (C_3_ of parent aromatic ring), δ 135.22 (C_1_ of substituent aromatic ring), δ 137.60 (C_1_ of parent aromatic ring), δ 151.24 (C_4_ of substituent aromatic ring), δ 151.22, 150.45 (C_2,5_ of parent aromatic ring), δ 152.66 (C_2_ of oxadiazol), δ 167.36 (-C=O); LC-MS (*m*/*z*): 511 (M + 1), molecular formula = C_21_H_17_ClF_6_N_2_O_4._

### 4.5. Anticancer Studies

#### 4.5.1. Cell Culture

Human glioblastoma LN229 cells (NCCS, Pune) were cultured in DMEM high glucose media (HiMedia, cat no: AL007S) supplemented with 10% heat-inactivated fetal bovine serum (FBS) (HiMedia, cat no: RM10432), and 1% Anti-bacterial and anti-mycotic solution (ABAM). Cells were maintained at 37 °C with 5% CO_2_ and growth media should be replenished every 2 days. Once the cells have attained 70–80% confluence, they were sub-cultured for cytotoxic studies. 

#### 4.5.2. Cytotoxic Assay

LN229 *glioblastoma* cells were grown in 48-well plates at a cell density of 0.03 × 10^6^ cells/well for the cytotoxicity experiment. After the cells have reached 70–80% confluence, they are treated with 1,3,4-Oxadiazolines derivatives, comprising 11 compounds at varying concentrations of 5 µM, 10 µM, 15 µM, 20 µM, and 25 µM. After 48 h of incubation at 37 °C in 5% CO_2_, the cell viability of LN229 was determined by MTT assay. In a brief, MTT (3-(4,5-dimethylthiazol-2-yl)-2,5-diphenyltetrazolium bromide) solution was added to each well at a final concentration of 0.5 mg/mL. MTT solution was removed after 4 h of incubation, and 300 μL of solubilizing buffer was then added to dissolve the purple crystal formazan forms [55,56]. The plates were shaken, and absorbance values are recorded at 570 nm. The results obtained from three simultaneous experiments are reported as mean ± standard deviation.

#### 4.5.3. Determination of IC_50_ Concentration for the Newly Synthesized Compounds

To determine the IC50 values for each compound, the absorbance values obtained for each treatment were subjected to non-linear regression analysis using graph pad prism software (version 8.3.0). Briefly, triplicate absorbance measurements from the MTT assay were imported into the XY sheet, the X values (concentration) were transformed to log, and the Y values (absorbance) were normalized to perform the nonlinear regression analysis. 

#### 4.5.4. Colony Formation Assay

The ability of a single cell to form a colony is the foundation of the clonogenic assay, sometimes referred to as the colony formation assay, which is an in vitro cell survival test. The ability of cancer cells to maintain reproductive integrity for an extended period was assessed in this test [57,58]. Here, LN229 cells were trypsinized and plated in a 24-well plate at even densities (50 cells/well). After allowing the cells to attach overnight, the media was replaced with 5% FBS media and treated with the screened compounds with a minimum and maximum of 10 µM and 25 µM (based on MTT results) and cultured at 37 °C. After 15 days of incubation, colonies were stained with 0.5% crystal violet and were manually counted and refined using an online application tool called “Promega colony counter”.

#### 4.5.5. Tunnel Assay

The efficacy of the compounds that were selected from the MTT assay was further evaluated using the tunel assay. This method is suitable for the detection of specific endonucleases that cleaves the genomic DNA between the nucleosomes during apoptosis. Cell apoptosis induced by the screened compounds was analyzed by using the Elabscience^®^ one-step TUNEL assay kit (catalog no: E-CK-A320). 

#### 4.5.6. Statistical Analysis

Graph Pad Prism 8.3.0 was used for analyzing all data and are analyzed using one-way ANOVA (and non-parametric or mixed). Data were presented as the mean standard deviation from a minimum of three independent experiments. *p* < 0.05.

### 4.6. Drosophila Stocks and Experimental Design

On wheat cream-agar medium, wild-type Canton-S (CS) and diabetic flies (Dilp2-Gal4/UAS-Dlip2-RNAi) were kept under 12:12 h of light and darkness and a temperature of 25 °C with relative humidity 65–70%. Male and female flies that were one day old were extracted and cultured for 10 days on food that had a consistent concentration (1 mM) of the chemicals **5b**, **5c**, **5d**, **5e**, **5f**, and **7k**. Twenty flies were taken, their haemolymph was collected, and the glucose concentration was calculated in accordance with the manufacturer’s instructions for a glucose assay kit that may be purchased (GAGO20, Sigma Aldrich). By utilizing the Bradford method at 595 nm and bovine serum albumin as the reference, the total protein content was calculated [59]. The glucose concentration was represented as µg of glucose/mg of protein. 

### 4.7. In Silico Studies

#### 4.7.1. Docking

Molecular docking study of 1,3,4-oxadiazole derivatives with Aurora Kinase A (AURKA, ID:1MQ4, 1.9 Å) and vascular endothelial growth factor receptor 2 (VEGFR-2, ID:4AG8, 1.95 Å) were carried out using Autodock Vina [60] to understand the protein–ligand interactions as anti-cancer inhibitors. Similarly, 1,3,4-oxadiazole derivatives were docked with α-Glucosidase (ID: 2QMJ, 1.9 Å) to understand the efficiency of synthesized molecules as anti-diabetic agents. The protein coordinate was prepared for the docking method using the Autodock GUI [42]. The protein was prepared by removing water and co-crystallized ligands. Gasteiger charges [61] were added to the protein. The 3D ligand structures were build and optimized by B3LYP level [62] and 6–31G(d,p) basis set [63] using Gaussian 09 package [64]. The protein is considered to be rigid during the docking procedure. Autogrid was employed to prepare a 3D grid with a pacing of 0.375 Å at the binding pocket [65]. The grid dimension was set to 70 Å × 64 Å × 64 Å and 60 Å × 64 Å × 60 Å at the binding pocket of AURKA and VEGFR-2 respectively. The grid dimension for the binding pocket of α-Glucosidase was set to 60 Å × 60 Å × 60 Å. The synthetic compounds that have the lowest binding energy and form stable protein–ligand complexes were regarded as promising therapeutic candidates. Further, the results were compared with known drugs; temozolomide and acarbose as standard for anti-cancer and anti-diabetic properties. Moreover, the synthesized compounds were subjected to ADMET prediction (absorption, distribution, metabolism, excretion, and toxicity) to understand the safety, efficacy, and bioavailability of a drug candidate using the SwissADME web tool (8). The drug-likeness of the compounds was determined using Lipinski’s rule of 5 (9).

#### 4.7.2. Molecular Dynamics Simulation Protocol

Molecular dynamics simulation is believed to be an accurate tool to predict the structural and dynamical properties of protein–ligand systems [66,67]. The most active ligand from the synthesized compounds based on the experimental observations and docking results were chosen to prepare the protein–ligand systems. Simulations were performed for AURKA/5b, VEGFR-2/5b, and α-Glucosidase/5d protein–ligand systems by GROMACS 2018.4 software [68] (3) using AMBER99SB-ILDN force field and TIP3P water model. The protein–ligand complex of AURKA, VEGFR-2, and α-Glucosidase was solvated in a periodic cubic box with a box length of 6.57 nm, 6.54 nm, and 10 nm respectively. The charge of the systems was neutralized, and energy minimization was performed using the steepest descent algorithm. NVT (at 300 K), and NPT (at 300 K and 1 atm) equilibration, each for 10 ns was carried out with a time step of 2 fs. The long-range electrostatic interactions were defined by the particle mesh Ewald method [69] and the short-range van der Waals cut-off was fixed to 1.2 nm. LINCS constraints [70] were used to restrain the bond involving hydrogen atoms. Finally, a production run was performed until the protein RMSD is converged.

## Data Availability

The data presented in this study are available in Appendix A.

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
