# Peer review of "2,5-Bis(2,2,2-trifluoroethoxy)phenyl-tethered 1,3,4-Oxadiazoles Derivatives: Synthesis, In Silico Studies, and Biological Assessment as Potential Candidates for Anti-Cancer and Anti-Diabetic Agent"

_molecules, 2022, doi:10.3390/molecules27248694_

Round 1

Reviewer 1 Report

Reviewer comments: 

The article by Sathyanarayana DS and colleagues discusses the synthesis, characterization, in-silico studies, and biological evaluation of some new 2,5-bis (2,2,2-trifluoroethoxy) phenyl-based 1,3,4-Oxadiazoles as potential candidates for Anti-cancer and Anti-diabetic. The manuscript is well written and explains the synthesis and characterization in-silico of Oxadiazoles as potential candidates for Anti-cancer and Anti-diabetic. However, the discussion of the results is also a significant concern. For example, the discussion is feeble in the sections on biological effects (cytotoxic effects, in vivo drosophila studies), whereas the conclusion is closer to the discussion.

Minor suggestions

·         Figure 2, in the description of results, the authors do not mention figure 2b, which similarly occurs with figure 3b

·         In the section ADME Toxicity Prediction, table 6 is missing.

·         Sections 2.2.2. Cytotoxic studies… The data about the cytotoxic effects of 5b, 5d, and 7m compounds are poorly discussed. The authors must discuss the possible reason for these results.

·         Figure 8A, 9A must be improved  

·         Please, define PC, NC in figure 9

·         Figure 10 shows the missing significance values that help assess the significant changes for each compound.

Author Response

Open Review-1

Reviewer comments:

The article by Sathyanarayana DS and colleagues discusses the synthesis, characterization, in-silico studies, and biological evaluation of some new 2,5-bis (2,2,2-trifluoroethoxy) phenyl-based 1,3,4-Oxadiazoles as potential candidates for Anti-cancer and Anti-diabetic. The manuscript is well written and explains the synthesis and characterization in-silico of Oxadiazoles as potential candidates for Anti-cancer and Anti-diabetic. However, the discussion of the results is also a significant concern. For example, the discussion is feeble in the sections on biological effects (cytotoxic effects, in vivo drosophila studies), whereas the conclusion is closer to the discussion.

Minor suggestions

  1. Figure 2, in the description of results, the authors do not mention figure 2b, which similarly occurs with figure 3b-

Response- We sincerely thank the reviewer. Suggestion  has been  incorporated in the revised manuscript.

  1. In the section ADME Toxicity Prediction, table 6 is missing.-

Response - We sincerely thank the reviewer. We do agree our mistake while writing the manuscript. Suggestion  has been  incorporated in the revised manuscript.

  1. Sections 2.2.2. Cytotoxic studies… The data about the cytotoxic effects of 5b, 5d, and 7m compre poorly discussed. The authors must discuss the possible reason for these results.-

Response - We sincerely thank the reviewer. Suggestion  has been  incorporated in the revised manuscript.

  1. Figure 8A, 9A must be improved -

Response - We sincerely thank the reviewer. Suggestion  has been  incorporated in the revised manuscript.

  1. Please, define PC, NC in figure 9 –

Response – We sincerely thank the reviewer. We do agree our mistake while writing the manuscript. Suggestion  has been  incorporated in the revised manuscript.

  1. Figure 10 shows the missing significance values that help assess the significant changes for each compound.-

Response - We once again thank the reviewer. We do agree our mistake while writing the manuscript. Suggestion  has been  incorporated in the revised manuscript.

Reviewer 2 Report

Sathyanarayana and group reported the synthesis of 2,5-bis (2,2,2-trifluoroethoxy) phenylbased 1,3,4-Oxadiazoles as potential candidates for Anti-cancer and Anti-diabetic. Results were well supported by computational studies and experimental data. The NMR of the compounds was clean. Most of the synthesized compounds were potent. This is overall good research and deserves to be published in this journal. There are some comments below authors need to address before publication

1.       Title of the paper was very lengthy and not very attractive. The authors should rewrite it.

2.       Add the word “atoms” in the second line after the sulfur word of the introduction part.

3.       Word anti-inflammatory was written two times. Please delete it.

4.       Authors should shuffle the last two paragraphs of the introduction part on page two e.g Diabetes mellitus introduction should be followed by the fruit fly rationale.

5.       Please add some references to the chemistry part as this was reported chemistry.

6.        Reaction conditions were incomplete in both schemes. e.g  temperature and reaction time were missing

7.       Authors really need to redraw the schemes using ChemDraw according to the journal guidelines.

8.       Authors should merge the two synthetic routes into one after intermediate 3 as the reaction conditions and common structure of the final compounds were the same.

9.       Ar groups should be presented in a tabular form in the scheme and need to remove from the scheme legend.

10.   Please remove the structural explanation of compound 7i from the result and discussion part as this information was already provided for compound 5a and does not need to repeat it.  

11.    Table 1 should be deleted as the same information was present in the experimental section.

12.   Authors can delete the docking figures of the inactive compounds.

13.   Structure-activity relationship needs to be improved as the authors did not discuss the activity profile of the most compounds

14.   Cell viability figures should be removed from the manuscript as these were also given in the supplementary information.

15.   IC50 value for the standard is significant which was missing in Table 7.

16.   Abbreviation IC50 should be consistent throughout the manuscript.

17.   Please concise the conclusion part.

18.   Reduce the NMR values up to two decimal places

19.   Remove the text from the NMR peaks.

20.   Most of the references were very old. Needs to update them.

21. Authors are advised to provide the rational design figure for the compounds.

Author Response

Open review -2

Sathyanarayana and group reported the synthesis of 2,5-bis (2,2,2-trifluoroethoxy) phenylbased 1,3,4-Oxadiazoles as potential candidates for Anti-cancer and Anti-diabetic. Results were well supported by computational studies and experimental data. The NMR of the compounds was clean. Most of the synthesized compounds were potent. This is overall good research and deserves to be published in this journal. There are some comments below authors need to address before publication

  1. Title of the paper was very lengthy and not very attractive. The authors should rewrite it.

Response – We thank the reviewer for the suggestions. We have addressed this and re written the title

  1. Add the word “atoms” in the second line after the sulfur word of the introduction part.

Response - We thank the reviewer for the suggestions. We have addressed this and suggestions has been incorporated in the revised manuscript.

  1. Word anti-inflammatory was written two times. Please delete it.

Response - We thank the reviewer for the suggestions. We have addressed this and suggestions has been incorporated in the revised manuscript.

  1. Authors should shuffle the last two paragraphs of the introduction part on page two e.g Diabetes mellitus introduction should be followed by the fruit fly rationale.

Response – We thank the reviewer for the suggestions. We have addressed this and suggestions has been incorporated in the revised manuscript.

  1. Please add some references to the chemistry part as this was reported chemistry.

Response – We sincerely thank the reviewer for the suggestions. Given  suggestions has been incorporated in the revised manuscript.

  1. Reaction conditions were incomplete in both schemes. e.g temperature and reaction time were missing

Response : We thank the reviewer for the suggestions. We have addressed this and suggestions has been incorporated in the revised manuscript.

  1. Authors really need to redraw the schemes using ChemDraw according to the journal guidelines.

Response : We thank the reviewer for the suggestions. We have addressed this and suggestions has been incorporated in the revised manuscript.

  1. Authors should merge the two synthetic routes into one after intermediate 3 as the reaction conditions and common structure of the final compounds were the same.

Response : We thank the reviewer for the suggestions. We have addressed this and suggestions has been incorporated in the revised manuscript.

  1. 9. Ar groups should be presented in a tabular form in the scheme and need to remove from the scheme legend.

Response – Authors once again thank the reviewer for this suggestions. Suggestion has been incorporated in the revised manuscript.

  1. Please remove the structural explanation of compound 7i from the result and discussion part as this information was already provided for compound 5a and does not need to repeat it.

Response – Authors  thank the reviewer for the suggestions. Suggestions has been incorporated in the revised manuscript

  1. Table 1 should be deleted as the same information was present in the experimental section.

Response – Authors  thank the reviewer for the suggestions. Given suggestion has been incorporated in the revised manuscript

  1. Authors can delete the docking figures of the inactive compounds.

Response – Authors  once again thank the reviewer for the suggestions. Suggestion has been incorporated in the revised manuscript.

  1. Structure-activity relationship needs to be improved as the authors did not discuss the activity profile of the most compounds

Response – We once again thank the reviewer for this suggestion & we have incorporated the same in the revised manuscript.

  1. Cell viability figures should be removed from the manuscript as these were also given in the supplementary information.

Response – We once again thank the reviewer for this suggestion & we have incorporated the same in the revised manuscript. Incorporated only in supplementary file.

  1. IC50 value for the standard is significant which was missing in Table 7.

Response:  We thank the reviewer for the suggestion, however, in this study we examined the maximum concentration of 25 µM in NIH3T3 normal fibroblast cells, which showed no significant cytotoxic effect in a normal cell line. Therefore, we didn’t analyse the IC50 for the control cell line. Also, we considered the standard (normal cell line) only to determine the maximum concentration that needed to be considered for the cytotoxic study. Notably, IC50 was calculated independently from the standard cell line; therefore, we didn’t give the standard IC50 values for standard. Concludingly, we are proposing that at 25 µM concentration, the screened compounds are not cytotoxic to normal cells, but they exhibit a cytotoxic effect in cancerous conditions. Therefore, IC50 values for standards are not mentioned in the table.  

  1. Abbreviation IC50 should be consistent throughout the manuscript.

Response – We once again thank the reviewer for this suggestion & we have incorporated the same in the revised manuscript.

  1. Please concise the conclusion part.

Response – We once again thank the reviewer for this suggestion & we have incorporated the same in the revised manuscript.

  1. Reduce the NMR values up to two decimal places

Response – We sincerely thank the reviewer. Suggestion  has been  incorporated in the revised manuscript.

  1. Remove the text from the NMR peaks.

Response – As per the suggestions, we have removed texts from proton NMR and 13C NMR. Retained only structure for easy identification.

  1. Most of the references were very old. Needs to update them.

Response – We once again thank the reviewer for this suggestion & we have incorporated the same in the revised manuscript.

  1. Authors are advised to provide the rational design figure for the compounds.

Response – We sincerely thank the reviewer for this suggestions. The required rational design figure requires molecular modeling/docking software. Unfortunately as our collaborator (one of the authors, who is specialized with theoretical modeling) had gone for vacation for about three weeks. In view of the same, we could not incorporated the same. However in all our future manuscripts, we will incorporate the same.

Round 2

Reviewer 1 Report

None